# Assembly landscape for the bacterial large ribosomal subunit

Kai Sheng[1], Ning Li [1], Jessica N. Rabuck-Gibbons[1,2], Xiyu Dong[1], Dmitry Lyumkis [1,2,3] & James R. Williamson [1] ✉

Assembly of ribosomes in bacteria is highly efficient, taking ~2-3 min, but this makes the abundance of assembly intermediates very low, which is a challenge for mechanistic understanding. Genetic perturbations of the assembly process create bottlenecks where intermediates accumulate, facilitating structural characterization. We use cryo-electron microscopy, with iterative sub-classification to identify intermediates in the assembly of the 50S ribosomal subunit from *E. coli*. The analysis of the ensemble of intermediates that spans the entire biogenesis pathway for the 50 S subunit was facilitated by a dimensionality reduction and cluster picking approach using PCA-UMAP-HDBSCAN. The identity of the cooperative folding units in the RNA with associated proteins is revealed, and the hierarchy of these units reveals a complete assembly map for all RNA and protein components. The assembly generally proceeds co-transcriptionally, with some flexibility in the landscape to ensure efficiency for this central cellular process under a variety of growth conditions.

Ribosome biogenesis is a complex but efficient process in rapidly growing bacteria. Assembly of a functional 70S ribosome completes in ~2–3 min[1,2] and involves the participation of 3 rRNAs, over 50 r-proteins, and dozens of assembly factors[3–6]. In vitro, reconstitution using various subsets of large subunit (50 S, LSU) proteins with rRNAs resulted in the Nierhaus assembly map[7–9], embodying the cooperativity and dependency for binding of LSU r-proteins to 23S rRNA. Critically absent from the Nierhaus map is the underlying folding of the rRNA that creates the binding sites for the r-proteins. In addition, the relationship of the observed cooperativity in vitro to the co-transcriptional assembly in cells remains to be determined.

The structure of the complete 50S subunit provides very few clues to the assembly pathway. The 23S rRNA secondary structure is organized into 6 domains based on phylogenetic analysis of secondary structure[10], but these domains are highly interdigitated in the complete subunit. Using a genetic depletion of the ribosomal large subunit protein bL17[11], a series of thirteen intermediates were identified using cryo-EM for the later stages of assembly, which was later expanded to

42 intermediates by further subclassification[12]. Analysis of this set of particles revealed a set of five assembly blocks[11], but the blocks did not specifically correspond to domains in the standard secondary structure[10,13]. In addition, the assembly blocks provided evidence for both parallel and sequential folding of RNA elements. The earliest intermediate that was discovered in the data revealed ordered density approximately encompassing the solvent side half of the 50S subunit. Genetic manipulation of assembly factors, such as SrmbB, ObgE, and RbgA[14–16], revealed additional details pertaining to the mechanism of late-stage assembly of the inter-subunit cleft of the 50S subunit. However, our understanding of LSU assembly is largely limited to the late stages. Furthermore, whether the principles of assembly are generalizable and how assembly pathways are related under diverse conditions of stress remains unclear.

DeaD is a cold shock protein in *E. coli*, with several annotated functions involving mRNA stability and association with the 50S ribosomal subunit[17–19]. The deletion strain Δ*deaD*, has a severe growth defect at low temperatures, and the sucrose gradient profile for this strain

[1]Department of Integrative Structural and Computational Biology, Department of Chemistry, and The Skaggs Institute for Chemical Biology, The Scripps Research Institute, La Jolla, CA 92037, USA. [2]Laboratory of Genetics, The Salk Institute for Biological Studies, La Jolla, CA 92037, USA. [3]Graduate School of Biological Sciences, Section of Molecular Biology, University of California San Diego, La Jolla, CA 92093, USA. ✉e-mail: jrwill@scripps.edu

shows significant accumulation of a pre-50S peak[17]. Cryo-EM analysis of the pre-50S peak from the Δ*deaD* strain revealed the earliest intermediate yet identified, consisting of domain I and three associated r-proteins. The analysis of this data was facilitated by an improved workflow for heterogeneous reconstruction using template-free ab-initio classification and iterative subclassification in CryoSPARC[20]. Further, we also developed a new toolbox of unsupervised feature extraction and electron density segmentation to identify assembly blocks based on single voxel behavior across a set of maps, which enables the subsequent cooperativity and dependency analysis.

Overall, we generated a set of 21 pre-50S density maps from the Δ*deaD* dataset, and we applied our segmentation and dependency analysis method to identify 10 cooperative assembly blocks. The set of blocks was organized into a block dependency map that demonstrated, for the first time, the integrated interdependency of the organization of rRNA helices and protein binding. The process by which the exit tunnel is formed was revealed during the assembly of the solvent half of the subunit, which then serves as a scaffold for 50S maturation. With the folding blocks of the entire subunit in hand, we revisited the previously reported bL17-depletion and Δ*srmB* datasets[11,15]. Remarkably, prior datasets are consistent with the block dependency derived from the Δ*deaD* dataset, which implies a unified early assembly pathway and a malleable maturation landscape in 50S biogenesis.

## Results

### Iterative ab-initio subclassification reveals new LSU assembly intermediates

The Δ*deaD* strain grown at 19 °C had a severe growth defect (Supplementary Table 1) that resulted in the accumulation of pre-50S particles in the sucrose gradient profile (Supplementary Fig. 1) The whole cell and pre-50S fraction proteomic data showed that a lack of DeaD at low-temperature results in a defect of ribosome assembly without alteration of r-protein expression (Supplementary Fig. 2a). The pre-50S fractions purified via sucrose gradient from the Δ*deaD* strain were subjected to quantitative proteomic mass spectrometry and RNA mass spectrometry analysis for RNA modification. The intermediates showed non-stoichiometric protein composition (Supplementary Fig. 3) and partial RNA modification (Supplementary Fig. 4), implying a heterogeneous composition of the assembly intermediates. Subsequently, single-particle cryo-EM data collection and analysis was applied using a similar approach to an iterative subclassification strategy previously reported[12]. We developed a robust implementation for iterative ab-initio subclassification, expanding on our previous approach[12], using CryoSPARC to identify heterogeneous populations of assembly 50S subunits, as described in the methods. Analysis using this protocol resulted in 21 distinct particle maps for the Δ*deaD* dataset, shown in Fig. 1a.

To compare with prior results, we re-analyzed two previously reported datasets using this updated protocol, one with intermediates from the depletion of bL17[11] and one with intermediates from a Δ*srmB* strain[15], resulting in 32 and 11 distinct particle maps, respectively, shown in Fig. 1b, c. The combined set of 64 maps was compared to identify similar classes of particles among the three individual sets of maps. The Euclidean distance matrix was calculated for the total set of 64 maps, thresholded at intensity 1.00. Agglomerative hierarchical clustering was performed using this matrix, which grouped the maps into 6 major classes (Fig. 1d), with the class distributions for the three datasets shown in Fig. 1e–g. Some of the new maps align well with the B, C, and E

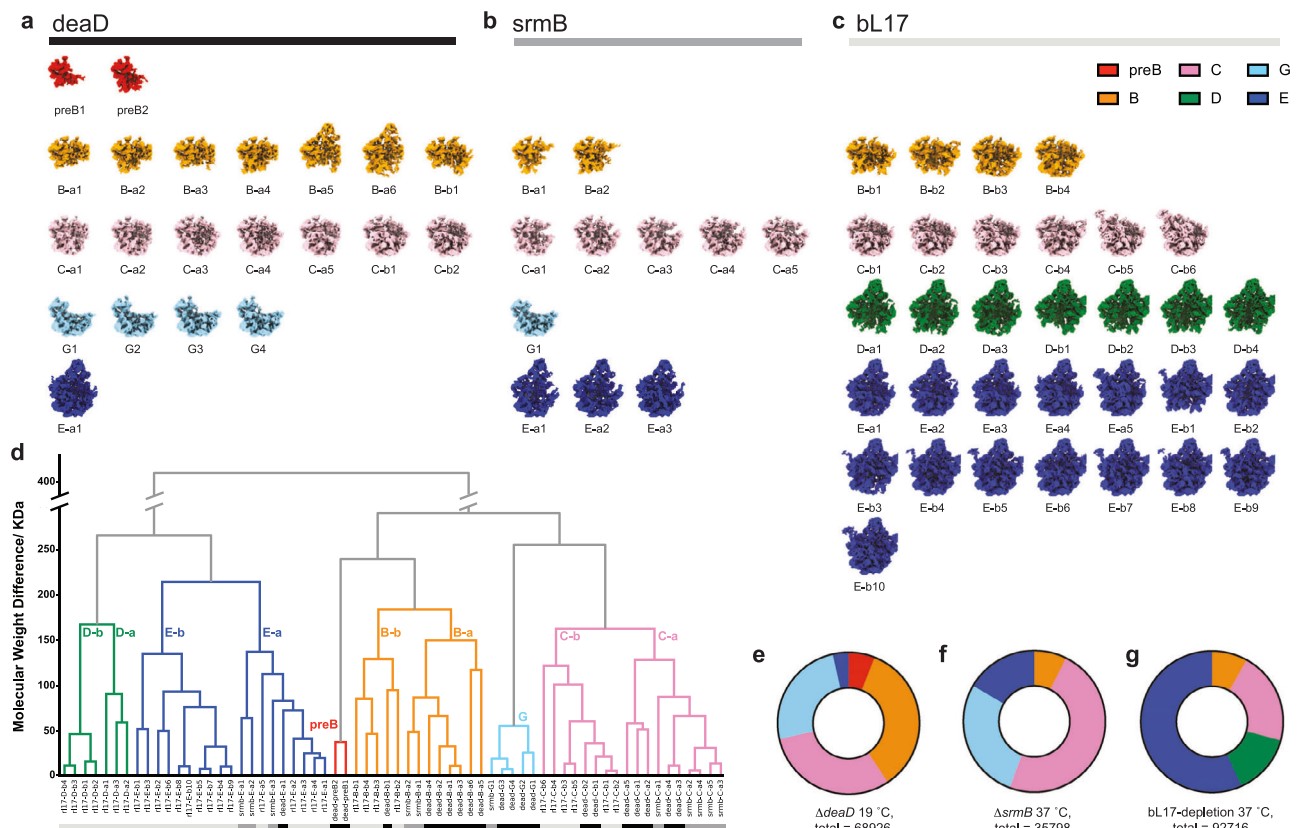

**Fig. 1 | Assembly intermediate density maps from three datasets.** Density maps reconstructed from **a** Δ*deaD*, **b** Δ*srmB*, and **c** bL17-depletion datasets, colored according to classes obtained from hierarchical analysis in (**d**). The Euclidean distance matrix, based on the molecular weight in kDa, was calculated among density maps, and the dendrogram resulting from hierarchical clustering is displayed, with the six main class branches colored accordingly. The bottom color bars are corresponding to (**a**–**c**), black = Δ*deaD*, dark gray = Δ*srmB*, and light gray = bL17-depletion. **e**–**g** Particle distribution among the main classes for the three datasets.

classes previously observed in the bL17 depletion strain[11,12], while the D-class is only observed in the bL17 dataset. Two new classes observed in the Δ*deaD* dataset were labeled as preB and G.

The newly discovered preB1 and preB2 classes, exclusively found in the Δ*deaD* dataset, represent the earliest intermediates observed among all in vivo studies of ribosome assembly, corresponding roughly to ~600 nucleotides of domain I and domain I + III of 23S rRNA, respectively. There is a new class related to the previous C class that is primarily observed in the Δ*deaD* dataset (25% of the particles in Fig. 1e, metadata in Supplementary Table 2), which we assign to the new G class that lacks a large portion of domain II helices, as well as uL13, bL20, and bL21. The Δ*deaD* particles mainly consist of the early B, C, and G classes. The E class, which is well-characterized in previous studies focusing on the late assembly process, only represents a small fraction of the particles. Thus, the Δ*deaD* dataset exhibited the largest breadth of early intermediates, ranging from the smallest intermediate yet observed to mature states and numerous in-between.

## Unsupervised voxel-based segmentation of density maps from Δ*deaD* intermediates reveals ten early assembly blocks

We developed a novel procedure to segment the maps using the dimensional reduction tools PCA[21] and UMAP[22], in combination with the HDBSCAN[23,24] algorithm for cluster identification (see Methods). Application of these tools to the Δ*deaD* cryo-EM dataset of LSU intermediates identified ten early assembly blocks as a basis set for the experimental maps (Fig. 2). Briefly, the set of 21 maps from Δ*deaD* was aligned and resampled to the same grid, thresholded at 99th percentile intensity, and the resulting 21 × 114,392 matrix of voxels with intensity above threshold in any dataset was used to generate 21 principal components. Plotting the first two principal components, PC1 and PC2, showed distinguishable features, but the clusters are not readily separated (Supplementary Fig. 5a). Some features can be extracted by thresholding a given PC at 1σ, such as the positive elements of PC2 as the base region or the negative elements of PC3 as domain I. (Supplementary Fig. 5b–d). In general, the desired contiguous density segments would be linear combinations of the PCs, but there is no straightforward method to solve for those segments. In addition, features in higher PCs are noisy and hard to identify, and voxels in these PCs cannot be unambiguously assigned to a single structural feature (Supplementary Fig. 6). Direct application of UMAP on the voxel intensities suffered from similar shortcomings in identifying contiguous segments (Supplementary Discussion 3, Supplementary Table 5). In contrast, subjecting the PCA reconstructed data to dimensionality reduction using UMAP gave rise to readily interpretable groups of voxels, as shown in Fig. 2a (Supplementary Fig. 7).

Finally, HDBSCAN was used to resolve and identify clusters of voxels in the UMAP representation in an unsupervised manner (Fig. 2a). The resulting clusters correspond to contiguous regions of density that serve as a basis set for the 21 maps from the Δ*deaD* dataset, as shown in Fig. 2b–k. The sequential application of PCA-UMAP-HDBSCAN analysis provides the cleanest, most intuitive segmentation of the set of density maps for 50S assembly, and this approach could be a powerful and general template for analyzing sets of maps from heterogeneous cryo-EM datasets.

To facilitate a comparison of the set of clusters emerging from the PCA-UMAP-HDBSCAN analysis, we first assigned numerical values to the occupancy of each cluster for a set of predefined RNA helices and r-protein volumes from the structure of the 50S subunit[11–13] (Fig. 2c). This occupancy analysis reveals that nearly all RNA and protein elements are uniquely assigned to a single cluster, confirming that these clusters can be used as a convenient basis set of assembly blocks. To understand the relationship among the newly defined blocks and to compare distinct features across individual maps, we performed a new occupancy analysis of the 21 intermediate maps based on the 10 basis blocks, resulting in a 10 × 21 matrix shown in Fig. 3a. The block names were based on prominent RNA or protein features, Figs. 2a–c and 3a.

## Mapping assembly block dependency for early 50S assembly

A folding block is operationally defined as a set of voxel intensities that are correlated across the 21 maps and are thus considered to be a cooperative assembly unit. The composition of the blocks is shown in Fig. 3a (Supplementary Fig. 7 and detailed discussion in SI), superimposed on the protein binding dependencies from the original Nierhaus map. The dependency among the blocks is evaluated by first making a scatterplot for each pair of rows of the occupancy matrix, then inferring the dependency using a quadrant analysis (Supplementary Fig. 8 and Methods). A similar approach was recently reported for a cryo-EM-based mechanistic study for bacterial small subunit assembly[25]. The uL1 base block was omitted from the dependency analysis, as it only occurred once in the Δ*deaD* dataset. After pruning the redundant edges in the dependency graph (Methods), the most direct dependencies between blocks are retained, and they are shown in colored bold arrows in Fig. 3b.

The assembly Core is a prerequisite for all subsequent assembly blocks and is the sole precursor to the consolidation of the bL20 or uL23 blocks, which constitute domain I–III, respectively, providing strong support for the natural 5′–3′ co-transcriptional direction of assembly. The uL3 block represents a significant portion of domain VI and can be organized by either the core + bL20 (in early B classes) or the core + uL23 (in G classes). We do not have evidence of direct folding of the uL3 on the assembly core, likely due to the rapid folding kinetics of the bL20 and uL23 blocks. The association of domains I–III

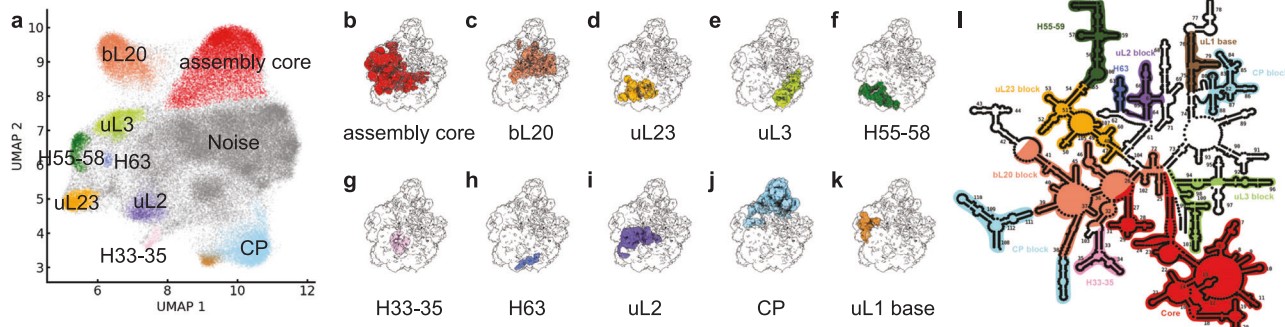

**Fig. 2 | Assembly blocks derived from segmentation using PCA-UMAP-HDBSCAN. a** Voxels above a 99-percentile threshold are well organized in UMAP space, with 10 contiguous volume blocks extracted by clustering with HDBSCAN colored (see Methods/SI). **b–k** Projections of the clusters into 3D space overlaid on the 50S subunit 99-percentile threshold mask (black outline), colored according to (**a**). **l** The 23S rRNA helices are colored according to the assembly blocks.

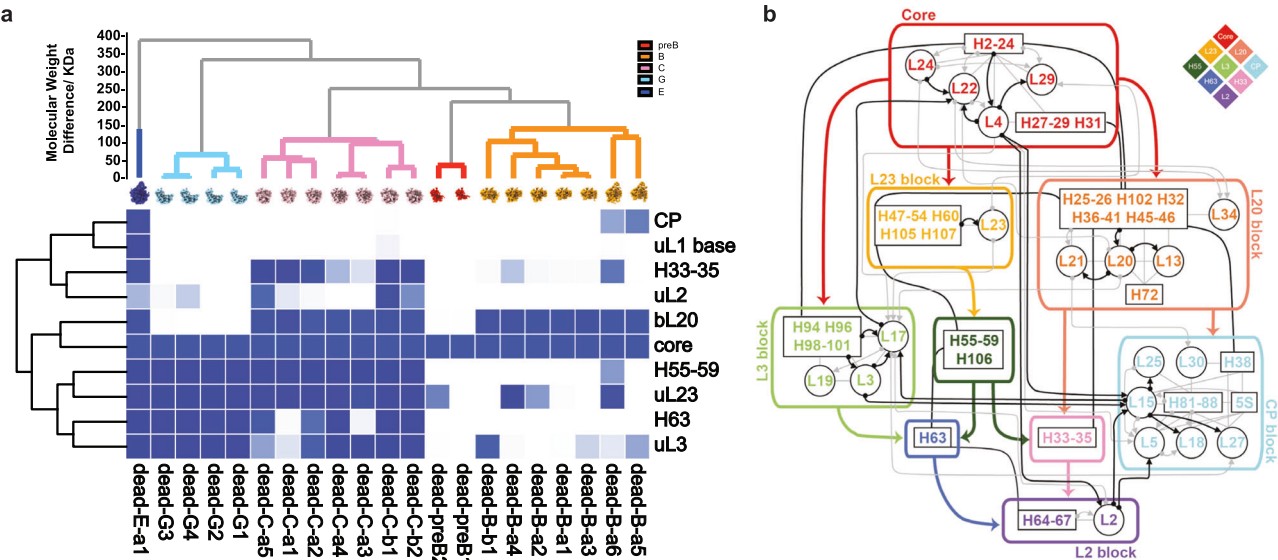

**Fig. 3 | Occupancy matrix of assembly blocks with the resulting block dependency map. a** Occupancy of 21 intermediate density maps from Δ*deaD* in terms of the 10 assembly blocks used for dependency analysis. **b** Block dependencies were determined using a quadrant analysis of the occupancy matrix in (**a**) (see SI). The blue color intensity ranges from white (no occupancy) to blue (full occupancy). All of the 23S and 5S rRNA helices are outlined in black boxes, with connections between elements in primary structure in solid black lines. Black/gray arrows show dependencies from the Nierhaus map as strong/weak interactions. The major block dependencies inferred from (**a**) are shown as bold-colored arrows. The diamond schematic diagram of the blocks, used in Fig. 4, is shown as an inset at the upper right.

and VI forms the majority of the solvent half of the 50 S subunit, and this group serves as the scaffold for the final assembly of the CP, the stalks, and ultimately the peptidyl transferase center (PTC). There are three small blocks consisting entirely of RNA, including the H55–59/106 block, which is part of domain III; the H63 block, which is part of domain IV; and the H33–35 block, which is part of domain II. Finally, there is a large block corresponding to the CP and the uL2 block which is part of domain IV. (Fig. 2l, detailed block descriptions in SI). The remaining parts of the 50 S subunit that are not represented by the 10 blocks correspond to the last folding steps forming the active site, which has been informed by previous work on the bL17 depletion strain. Moreover, the newly identified assembly blocks subdivided previously reported assembly, which illuminated key features of early 50S assembly (Supplementary Fig. 9).

### Placing RNA helices in the early stages of the assembly map

The comprehensive assembly map shown in Fig. 3b includes both the r-proteins and the rRNA helices defined in the secondary structure. The well-known Nierhaus assembly map established the basis for thermodynamic cooperativity among the LSU proteins binding to the 23 S rRNA[7–9]. For the first time, it is possible to intertwine RNA secondary structure elements with the LSU proteins to produce an RNA-protein assembly map. This is particularly revealing for the earliest stages of assembly, and it is now clear that assembly primarily proceeds in the 5′−3′ direction, consistent with a co-transcriptional organization of the folding blocks[26].

### The minimal requirement for CP formation

The dependency graph in Fig. 3b shows that the minimal requirement for CP formation is the assembly core docked with the bL20 block. The deaD-B-a5 particle, composed of the assembly core, the bL20 block, and a partially formed uL3 block, is the smallest intermediate containing ordered density for the CP. The volume for deaD-B-a6 is larger than for deaD-B-a5 where the uL3 block is not formed at all. It appears that the CP formation only requires the assembly core and bL20 block, which is consistent with the dependency graph. The bL20 block represents a continuous primary sequence from domain I

to domain II except for H38cp and H42–44 (base for L7/12 stalk). Once the H38bd is formed, the H38cp will recruit corresponding proteins, 5S, and part of domain V (H81–88) and form an intact CP (Supplementary Fig. 10d).

### Formation of exit tunnel in early intermediates

The discovery of early intermediates reveals the layer-by-layer formation of the exit tunnel from the solvent side towards the inter-subunit side, which finally forms the PTC. We split the exit tunnel (ET) into two parts: $ET_{solvent}$ and $ET_{PTC}$ (Supplementary Fig. 11). It is more relevant to discuss the $ET_{PTC}$ formation in the bL17 depletion dataset since they contain more mature structures, while the less mature set of Δ*deaD* intermediates, allows a focus on the $ET_{solvent}$ formation. By analyzing the structure of the secM and vemP peptide trapped on a translating ribosome[27,28], we generated a list of 50 S contacts in each of the assembly blocks within 5 Å of the trapped peptide (Supplementary Table 3).

From the assembly blocks, we can easily assign the resolved residues of $ET_{solvent}$ into the assembly core, bL20, uL23 block, and H33-35 block. Intermediates that do not contain all three of these blocks, namely the preB, G and B (except deaD-B-a6) class and part of the early C class, do not have a fully structured $ET_{solvent}$. In the dependency graph, we know the formation of H33-35 is dependent on the bL20 block and H55–58, which implies that H55–58 is also a prerequisite for $ET_{solvent}$ formation. For example, though the matured base region formed in the G class intermediates, none have the bL20 block formed, and so they never have the H33–35 block formed, resulting in incomplete $ET_{solvent}$ formation. All of the C classes have the H55–59, uL23 block, and bL20 blocks, and they only need the H33–35 formation to complete the $ET_{solvent}$ during the maturation.

These results differ significantly from the Steinberg analysis of the evolution of the 23 S rRNA from the primordial PTC[29], in which domain I (assembly core), domain II (bL20 block), and domain III (uL23 block) form after PTC formation, in the evolutionary sense. Presumably, the folding of the proto-ribosome was organized around the PTC, and there was likely an important stage in evolution after insertion of the domains into the proto-PTC, where the assembly process was

reorganized from forming the PTC first to forming the ET$_{solvent}$ first as a scaffold on which to build the PTC.

## Different perturbations reshape the ribosome assembly landscape

Given the observed dependencies in Fig. 3b, a set of 29 possible combinations of allowable structures can be enumerated, and of these, 14 were observed in the ΔdeaD dataset. These structures can be organized into a putative assembly pathway, connecting similar structures according to the dependencies, as shown in Fig. 4a. We also calculated the block occupancy for ΔsrmB and bL17 datasets (Supplementary Fig. 12). With the occupancy matrix, we validate

the quadrant analysis across the three datasets, using ΔdeaD thresholding criteria (Supplementary Fig. 13). The quadrant analysis matched the ΔdeaD results. Across the ΔdeaD, ΔsrmB, and bL17 datasets, 21 of the 29 possible structures are observed, and there are no combinations that violate the dependencies based on the ΔdeaD data alone (Fig. 3b). Comparing different datasets on the pathway, the ΔdeaD intermediates are distributed earlier than the other two, and there are few intermediates containing CP blocks.

The ΔsrmB and bL17 datasets populate intermediates not observed in the ΔdeaD dataset, but the dependencies are consistent with those observed within the ΔdeaD dataset, which implies a

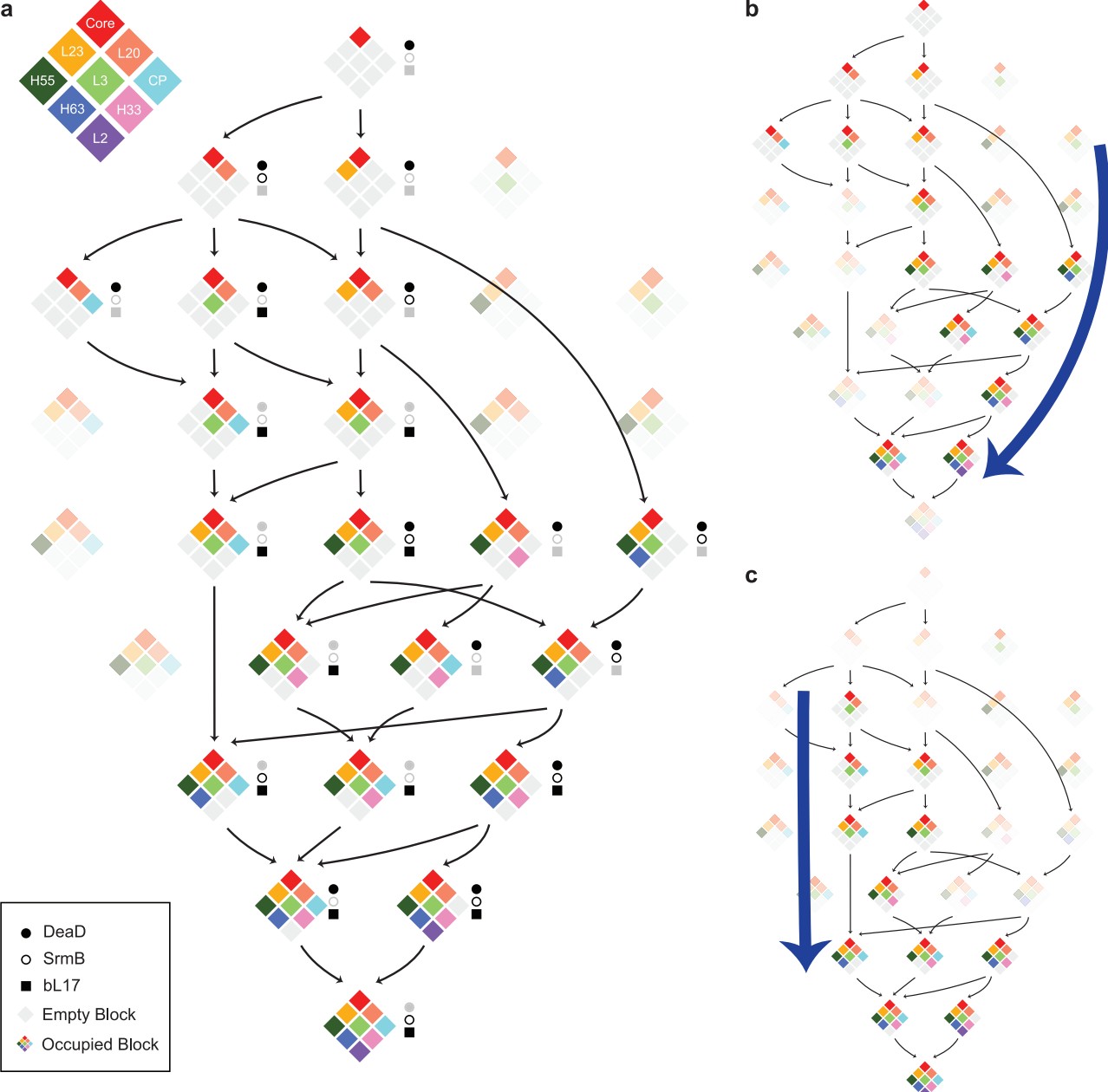

**Fig. 4 | Schematic comparison of assembly pathways. a** From the block dependencies in Fig. 3b, 29 possible intermediates are arranged from top to bottom, based on increasing block number. Intermediates observed within the datasets are shown in color, and unoccupied blocks are shown in gray. The presence of an intermediate in ΔdeaD, ΔsrmB, or bL17-depletion datasets is indicated by black closed circles, open circles, and squares, respectively. There are eight combinations

consistent with the block dependencies that are not observed, indicated by faded intensities. Arrows connect the nearest precursors, with disassembly not allowed, requiring the assembly core as the parent node. Display of intermediate found in ΔdeaD (**b**) and bL17-depletion (**c**) datasets. Blue bold arrows highlight the change in flux through the same set of intermediates in the different datasets. (See SI for the ΔsrmB intermediate pathway, which is similar to ΔdeaD).

universal block-wise parallel pathway for the assembly mechanism (Fig. 4a). Interestingly, the Δ*deaD* and Δ*srmB* datasets share a similar assembly path that proceeds through the unique G class, which is not observed in the bL17-depletion strain. We showed that Δ*srmB* intermediates are also depleted of uL13, resulting in a defect in the bL20 block formation. Since the bL20 block is essential for the formation of the CP block, in the Δ*srmB* dataset, there are accordingly fewer intermediates with the CP block formed at the very beginning, flowing through the right side of the landscape in Fig. 4b (Δ*srmB* pathway in Supplementary Fig. 14). In contrast, bL17 is at the bottom of the mature 50 S and is thought to play an important role in the blocks of the base region (uL2, uL3 block, H33–35 and H63 block), so the bL17-depletion intermediates have defects in the base blocks' formation, the CP forms earlier intermediates that accumulate with an incomplete base region, and mostly transition through the left part of the assembly landscape in Fig. 4c, which led to unique D classes in the bL17-depletion assembly pathway. Although there is no knowledge of the substrate for DeaD during ribosome assembly, the Δ*deaD* strain shares many intermediates with the Δ*srmB* strain, which implies the two helicases operate on substrates that are present at similar times during assembly.

## Continuous learning of 50S folding dependency with new datasets

We have shown that the essential features of the early stage 50 S assembly pathways for datasets from different perturbations can be represented using segments derived from the Δ*deaD* dataset alone. As the field progresses, additional information will become available through novel perturbations and improved methods for data analysis, and it becomes interesting to consider how the segmentation analysis differs when additional datasets are included. Performing the PCA-UMAP-HDBSCAN analysis on the three combined datasets with 64 intermediate density maps revealed a set of 18 assembly blocks (namely, blk01 to blk18, Supplementary Fig. 15, Supplementary Table 4). To compare to the 10 Δ*deaD* blocks, occupancy analysis was performed for the new set of blocks (Supplementary Fig. 16), revealing 8 blocks that were perfectly aligned in common. (Supplementary Fig. 17a–d, f–h). Interestingly, a small segment was separated from the bL20 block from Δ*deaD* that corresponded to uL13 and H25 (Supplementary Fig. 17e). The depletion of uL13 was a feature of intermediates in the Δ*srmB* strain, implying that the site of action of SrmB may be related to H25. In a similar manner, bL17 was segmented out from the uL3 block owing to including bL17-depletion dataset (Supplementary Fig. 17i). Further, new blocks were identified in the 3-dataset analysis, such as H67–69 and the uL10/uL11 stalks (Supplementary Fig. 18c, d), that are important in late assembly stages and are not present in the early intermediates from Δ*deaD* datasets. The blk14 even includes a non-native density for assembly factor YjgA (Supplementary Fig. 18g), which is also a feature of a subset of bL17-depletion intermediates, capturing the distinct dumbbell shape of the assembly factor surrounded by bL31 and H74/80/93 as a docking site for YjgA. The PCA-UMAP-HDBSCAN approach can robustly identify distinct and common features, for both native and non-native density, in a complex assembly landscape and provides for continuous learning of the ribosome biogenesis pathway with increased spatiotemporal resolution as more observations are included using different perturbations.

## Discussion

Through our analysis of ribosome assembly within a bacterial strain perturbed by the deletion of Δ*deaD* and low temperature, we identified a diverse set of intermediates that span the entire pathway for assembly of the 50 S subunit in vivo. The series of intermediates discovered in the dataset include the earliest particles composed roughly of domain I at the 5′ ends of the subunit, proceeding with the assembly of the solvent portion of the peptide exit tunnel prior to assembly of the inter-subunit face and PTC.

To systematically investigate the assembly process in a relatively unbiased way, we developed a novel quantitative segmentation tool that can be readily applied to a set of density maps and used the resulting segments to develop an assembly pathway for the entire subunit. In contrast to arbitrary definitions of structural elements, particularly rRNA, based on primary or secondary structure, the assembly blocks generated by PCA-UMAP were exclusively based on the dataset itself. This approach resulted in a more precise set of structural elements throughout the 50S biogenesis pathway. This segmentation can be readily applied to other datasets resulting from different perturbations, providing a unifying set of intermediates across datasets with differing fluxes through the pathway.

These data and the subsequent analysis provide a comprehensive view of the overall assembly of the 50 S subunit that integrates a significant body of data from decades of research into a coherent assembly map containing all r-proteins and RNA helical elements. This is the first time that the assembly dependency of helices, both within the same domain and across different domains, has been demonstrated through tertiary interactions. For instance, the formation of H33–35 requires not only the other majority of domain II helices in bL20 blocks but also H55–59 and H106 in domain III. Additionally, the existence of H55–59 and H106 is necessary for the formation of H63 from domain IV.

There are similarities and important distinctions to the nucleolar pre-60S intermediates identified in yeast[30,31]. The earliest yeast intermediate identified corresponds to domains I/II, onto which either domain III or VI can assemble. These intermediates correspond roughly to the core/L20 block followed by the uL23 block or uL3 block, respectively, in the present work. In addition to these states, we observe earlier intermediates corresponding to the domain I alone (Core) and domains I/III (Core/uL23 block) in comparison to yeast. Subsequent addition of domains occurs in a different order, with domain V assembling prior to domain IV in yeast, while the reverse is observed in the bacterial work where the active site in domain V is always the last to form. Perhaps most striking is the nearly complete absence of assembly factors in the present set of intermediates, compared to over 20 clearly resolved factors in the yeast intermediate structures. The reasons for this are not entirely clear, but this observation implies that the many known factors in bacteria must be transiently associated with the intermediates in such a way that they are not kinetically stable for purification. Overall, the rough correspondence of the domain assembly order is consistent with the sequence and functional conservation of ribosomes between kingdoms, but many details have been re-engineered during evolution.

Two other studies have recently appeared describing a similar range of intermediates from very different experimental approaches. Intermediates resulting from time points in an in vitro reconstitution of *E. coli* 50S subunits were recently identified, showing a remarkably similar range of structures to the present work[32]. The implied rough order of domain assembly is Domain I followed by either II/VI or III/VI, and the earliest domain I intermediate is smaller by one protein and a helix, representing the earliest intermediate observed thus far[32]. That work is particularly important due to its connection to the significant body of work on protein dependence in in vitro reconstitution[33]. Our laboratory recently published a similar set of intermediates resulting from co-transcriptional ribosome synthesis using the iSAT system[34]. Perhaps what is most remarkable is that very similar overall pathways emerge from three entirely different systems: in vitro reconstitution[32], iSAT[34], and cellular intermediates, in the present work. There are many interesting differences among the datasets that provide a striking level of detail on this complex assembly process. Nevertheless, the concordance provides strong validation that the observations are

pertinent to the assembly process and not an artifact of any particular experimental approach.

In the analysis of the Δ*deaD* intermediates accumulated at low temperatures, we have developed a novel segmentation method for the analysis of related sets of electron density maps, and we have used the segmentation to develop a hierarchy of assembly steps that are embodied by the set of maps and provide a putative mechanistic order for assembly. Further, this analysis has proven to be applicable to other datasets, and the dependencies observed in the Δ*deaD* data set are consistent with intermediates observed in other independent datasets, even though they are not observed in the Δ*deaD* data set. This approach should prove to be powerful as a platform moving forward to integrate mechanistic information as the as-yet mysterious roles of the bacterial assembly factors are elucidated.

## Methods

### Bacterial strains and plasmid construction
Strains BW25113 *E. coli* (WT) and BW25113 (Δ*deaD*) from the Keio Knockout Collection were purchased from the E.coli Genetic Stock Center[35]. The pHSL-*deaD*, homoserine lactone (HSL) -inducible DeaD expression plasmid was generated by Gibson cloning from pHSL-*rplQ*[11], replacing the coding region of *rplQ* with *deaD* coding sequence in pHSL. The Δ*deaD*-pHSL-*deaD* strain was obtained by transformation of pHSL-*deaD* into strain Δ*deaD*.

### Cell growth and sucrose gradient purification for ribosome particles
WT, Δ*deaD*, and Δ*deaD*-pHSL-*deaD* strains were inoculated in LB medium and grown overnight, then diluted into fresh LB at 20 °C. Either 0 or 2.5 nM HSL was added into the Δ*deaD*-pHSL-*deaD* strain during cell culture. Cells were harvested at OD600 -0.4 by centrifugation at 4000×*g* for 15 min, followed by lysis in Buffer A (20 mM Tris-HCl pH 7.5, 100 mM NH$_4$Cl, 10 mM MgCl$_2$, 0.5 mM EDTA, 6 mM β-mercaptoethanol) and 20 U/ml DNase I (Sigma) by a mini bead beater using 0.1-mm zirconia/silica beads (3 × 60 s pulses with 1 min on ice in between). Insoluble cell debris and beads were then removed by two centrifugation steps: 31,000×*g* for 10 min, transferring the supernatant to a new tube, and then again 31,000×*g* for 90 min. The clarified cell lysates (10 A$_{260}$ units) were loaded onto a 33 mL 10–40 % w/v sucrose gradient (50 mM Tris-HCl 7.8, 100 mM NH$_4$Cl, 10 mM MgCl$_2$, 6 mM β-mercaptoethanol) then centrifuged in a Beckman SW32 rotor at 80,000×*g* for 16 h at 4 °C. Gradients were fractionated using a Brandel gradient fractionator. Based on the UV 254 nm trace, gradient fractions corresponding to the pre-50S peak were collected and combined. To prepare the fractions for cryo-EM analysis, 3× volumes of buffer A were added prior to concentration in a 100 kDa cutoff concentrator (Amicon) 3 times to eliminate sucrose and to equilibrate to buffer A.

### Cryo-EM sample preparation and data collection
The purified pre-50S sample was diluted to 0.6 mg/ml with buffer A, and 3 μL of the sample was applied to a plasma-cleaned gold grid in the cold room. Grids were manually frozen in liquid ethane, and single-particle cryo-EM data was collected on a Thermo Fisher Scientific Titan Krios electron microscope operating at 300 keV equipped with a Gatan K2 Summit detector using the Leginon software[36], with a pixel size of 1.31 Å at 22,500× magnification. A dose of 33–35 e/Å$^2$ across 50 frames was used for a dose rate of ~5.8 e/pix/s. To overcome problems of the preferred orientation of particles on the grid and facilitate image classification, data was collected using a tilt angle of −20°[37]. A total of 1031 micrographs were collected.

### Electron microscopy micrograph processing
Data pre-processing, including motion correction and CTF estimation, was performed within the Appion pipeline[38]. Frames were aligned using MotionCor2[39], and the contrast transfer function (CTF) for all micrographs was estimated with CTFFind4.1[40]. The aligned frame sums were then imported into Relion. A total of 322,187 particles from the dataset were picked with auto-picking in RELION3. The particles were extracted in a 160 × 160 × 160 box with a bin of 2. Next, 2D classification and manual class curation were used to remove 30 S subunits and 70 S ribosomes, as well as other spurious particles that clearly did not belong to assembling LSUs. The curated particles from selected classes were further cleaned with 3D classification using a C-class 50 S intermediate as a template[12]. After 3D classification from RELION, classes that did not produce an interpretable map were eliminated. The resulting 273,729 particles were exported and analyzed in CryoSPARC3[20]. Particle stacks for bL17-depletion and Δ*srmB* were prepared using the same procedure on previously acquired micrographs[11,15], resulting in 123,804 and 273,620 particles, respectively.

### Iterative classification with ab-initio reconstruction in CryoSPARC and hierarchical analysis
For each dataset, the resulting particle stacks were imported to CryoSPARC and directly subjected to ab-initio reconstruction, requesting for 4 classes using default parameters. Each resulting interpretable class (Supplementary Fig. 19) was subjected to another round of ab-initio reconstruction using the same parameters. This procedure was performed iteratively until the particle number in a class was less than 2000, in which case the 3D reconstruction would result in low-resolution maps. All reconstructions with fewer than 2000 particles were subjected to ab-initio reconstruction, requesting 1 class prior to 3D refinement in CryoSPARC. An example of this workflow, and the resulting reconstruction, is shown in SI for the Δ*deaD* dataset. All refined density maps were aligned and resampled to the same 50S ribosome reference (bL17-depletion dataset E) in ChimeraX. All maps with a resolution below 10 Å were discarded, and the remaining resampled density maps were thresholded at intensity 1.00. Pairwise difference maps were calculated for the binarized (thresholded at 1.00) maps, the sum of the difference map A-B and difference map B-A for hierarchical clustering using the Ward linkage. Maps were displayed with the resulting dendrogram, and pairs of maps with a difference of <10 kDa were merged into one class. The value of 10 kDa was defined previously as a valid merging criterion, as it represents the average molecular weight of all proteins and rRNA helices constituting the LSU[12]. This step is important as similar classes can emerge from hiding at various stages of the iterative subclassification[12]. The merged particles were next subjected to an ab-initio reconstruction and 3D refinement to produce the final map for the class. (Supplementary Data 1, Supplementary Fig. 20) Finally, a hierarchical clustering analysis was performed across all three datasets in the same way to allow a ready comparison of maps from the different datasets.

### Segmentation using PCA-UMAP-HDBSCAN with Δ*deaD* intermediates and three datasets
The 21 resampled Δ*deaD* intermediate maps were thresholded at 99 percentile intensity, and the set of 114,392 voxels with nonzero intensity in at least one map resulted in a 21 × 114,392 intensity array. Principle component analysis (PCA in Scikit-Learn)[21,41] was performed on this array, giving PCA transformed matrix of the same dimensions. UMAP[22] analysis was performed on the PCA matrix using 2 components with 100 nearest neighbors using the Canberra metric, resulting in a 2 × 114,392 matrix, projecting each voxel above the intensity threshold into a UMAP$_{1,2}$ space. The hyperparameters, including distance metrics and numbers of nearest neighbors and PC inputs for UMAP, are carefully explored (Supplementary Figs. 21–25, see detailed discussion in SI). The combination of PCA-UMAP compared to UMAP only showed better stability (Supplementary Fig. 26). In the UMAP$_{1,2}$

space, HDBSCAN (min_cluster_size = 100, min_samples = 100)[23,24] was performed to assign voxels to individual clusters, resulting 9 blocks representing contiguous regions of density in Cartesian space, and one noise block. The 9 blocks were considered assembly blocks, corresponding to a basis set of voxels that have correlated intensities in the input set of 21 maps. Blocks were named according to salient structural or compositional features. The CP and uL1 base blocks were further separated with another round of HDBSCAN (min_cluster_size = 10, min_samples = 10). All resulting assembly blocks were cleaned by dust filtering in ChimeraX[42] prior to use in occupancy analysis. (See SI for algorithm parameters discussion). Similarly, 64 intermediates from three datasets generated a 64 × 140,545 matrix. PCA and UMAP were performed direct on the first dimension of the matrix. HDBSCAN (min_cluster_size = 200, min_samples = 10) was performed to extract 18 assembly blocks.

### Occupancy and dependency analysis

The Occupancy of each assembly block was calculated for each of the density maps thresholded at intensity 1.00. Briefly, the number of voxels above the threshold is counted in each block and then normalized to the total number of voxels in the block. The occupied fraction for each block is then normalized to the core block occupancy in each density map.

The dependency between any pair of blocks ($i,j$) was obtained by quadrant analysis of a scatter plot of the occupancy for block $i$ on the x-axis and block $j$ on the y-axis (Supplementary Fig. 4a). The dashed binarization lines for the horizontal and vertical directions were calculated by the following Eq. (1),

$$\text{Binarization line} = \text{mean (smallest occupancy values above threshold line} \\ + \text{largest occupancy values below threshold line)}$$

$$(1)$$

\* threshold line is defined in Supplementary Fig. 27

The $x$ and $y$ binarization lines divide the scatter plot into four quadrants: QI = lower left, QII = lower right, QIII = upper left, QIV = upper right. To infer the relationship between block $i$ and $j$, the number of points in each quadrant was counted for the scatter plot. The relationship between block $i$ and $j$ falls into one of three scenarios (Supplementary Fig. 4a). With points only in QI and/or QIV, blocks $i$ and $j$ are correlated. With dots in both QII/QIII, blocks $i$ and $j$ are not correlated. With dots only in QI/QII/QIV or QII/QIV, block $j$ should depend on block $j$ (red scatter plots in Supplementary Fig. 4b). With dots in only QI/QIII/QIV or QIII/QIV, block $i$ should depend on block $j$. (blue scatter plots in Supplementary Fig. 4b). If block $i$ depends on block $j$, an arrow from $j$ to $i$ will be drawn in the dependency map. The comprehensive dependency plot is now ready for pruning with defined rules (Supplementary Fig. 28) with network package[43].

### Reporting summary

Further information on research design is available in the Nature Portfolio Reporting Summary linked to this article.

### Data availability

The data supporting the findings of this study are available from the corresponding authors upon reasonable request. Processed quantitative mass spectrometry data for RNA and protein are included in Supplementary Data 2. All 64 electron density maps for pre-50S intermediates were deposited on EMDB, dead-B-a1[https://www.ebi.ac.uk/emdb/EMD-40517], dead-B-a2[https://www.ebi.ac.uk/emdb/EMD-40519], dead-B-a3 [https://www.ebi.ac.uk/emdb/EMD-40520], dead-B-a4[https://www.ebi.ac.uk/emdb/EMD-40524], dead-B-a5[https://www.ebi.ac.uk/emdb/EMD-40526], dead-B-a6[https://www.ebi.ac.uk/emdb/EMD-40528], dead-B-b1 [https://www.ebi.ac.uk/emdb/EMD-40530], dead-C-a1[https://www.ebi.

ac.uk/emdb/EMD-40532], dead-C-a2[https://www.ebi.ac.uk/emdb/EMD-40534], dead-C-a3[https://www.ebi.ac.uk/emdb/EMD-40536], dead-C-a4 [https://www.ebi.ac.uk/emdb/EMD-40538], dead-C-a5[https://www.ebi.ac.uk/emdb/EMD-40540], dead-C-b1[https://www.ebi.ac.uk/emdb/EMD-40542], dead-C-b2[https://www.ebi.ac.uk/emdb/EMD-40544], dead-E-a1 [https://www.ebi.ac.uk/emdb/EMD-40546], dead-G1[https://www.ebi.ac.uk/emdb/EMD-40548], dead-G2[https://www.ebi.ac.uk/emdb/EMD-40550], dead-G3[https://www.ebi.ac.uk/emdb/EMD-40552], dead-G4[https://www.ebi.ac.uk/emdb/EMD-40555], dead-preB1[https://www.ebi.ac.uk/emdb/EMD-40551], dead-preB2[https://www.ebi.ac.uk/emdb/EMD-40549], rl17-B-b1[https://www.ebi.ac.uk/emdb/EMD-40309], rl17-B-b2[https://www.ebi.ac.uk/emdb/EMD-40311], rl17-B-b3[https://www.ebi.ac.uk/emdb/EMD-40313], rl17-B-b4[https://www.ebi.ac.uk/emdb/EMD-40511], rl17-C-b1[https://www.ebi.ac.uk/emdb/EMD-40314], rl17-C-b2[https://www.ebi.ac.uk/emdb/EMD-40315], rl17-C-b3[https://www.ebi.ac.uk/emdb/EMD-40317], rl17-C-b4[https://www.ebi.ac.uk/emdb/EMD-40319], rl17-C-b5[https://www.ebi.ac.uk/emdb/EMD-40321], rl17-C-b6[https://www.ebi.ac.uk/emdb/EMD-40323], rl17-D-a1[https://www.ebi.ac.uk/emdb/EMD-40327], rl17-D-a2[https://www.ebi.ac.uk/emdb/EMD-40329], rl17-D-a3[https://www.ebi.ac.uk/emdb/EMD-40331], rl17-D-b1[https://www.ebi.ac.uk/emdb/EMD-40333], rl17-D-b2[https://www.ebi.ac.uk/emdb/EMD-40512], rl17-D-b3[https://www.ebi.ac.uk/emdb/EMD-40514], rl17-D-b4[https://www.ebi.ac.uk/emdb/EMD-40516], rl17-E-a1[https://www.ebi.ac.uk/emdb/EMD-40518], rl17-E-a2[https://www.ebi.ac.uk/emdb/EMD-40521], rl17-E-a3[https://www.ebi.ac.uk/emdb/EMD-40523], rl17-E-a4[https://www.ebi.ac.uk/emdb/EMD-40525], rl17-E-a5[https://www.ebi.ac.uk/emdb/EMD-40527], rl17-E-b1[https://www.ebi.ac.uk/emdb/EMD-40529], rl17-E-b10[https://www.ebi.ac.uk/emdb/EMD-40531], rl17-E-b2[https://www.ebi.ac.uk/emdb/EMD-40533], rl17-E-b3[https://www.ebi.ac.uk/emdb/EMD-40535], rl17-E-b4[https://www.ebi.ac.uk/emdb/EMD-40537], rl17-E-b5[https://www.ebi.ac.uk/emdb/EMD-40539], rl17-E-b6[https://www.ebi.ac.uk/emdb/EMD-40541], rl17-E-b7[https://www.ebi.ac.uk/emdb/EMD-40543], rl17-E-b8[https://www.ebi.ac.uk/emdb/EMD-40545], rl17-E-b9[https://www.ebi.ac.uk/emdb/EMD-40547], srmb-B-a1[https://www.ebi.ac.uk/emdb/EMD-40316], srmb-B-a2[https://www.ebi.ac.uk/emdb/EMD-40318], srmb-C-a1[https://www.ebi.ac.uk/emdb/EMD-40320], srmb-C-a2[https://www.ebi.ac.uk/emdb/EMD-40322], srmb-C-a3[https://www.ebi.ac.uk/emdb/EMD-40324], srmb-C-a4[https://www.ebi.ac.uk/emdb/EMD-40325], srmb-C-a5[https://www.ebi.ac.uk/emdb/EMD-40328], srmb-E-a1[https://www.ebi.ac.uk/emdb/EMD-40330], srmb-E-a2[https://www.ebi.ac.uk/emdb/EMD-40332], srmb-E-a3[https://www.ebi.ac.uk/emdb/EMD-40513], srmb-G1[https://www.ebi.ac.uk/emdb/EMD-40515]. Detailed information is included in the supplementary information (Supplementary Table 2).

### Code availability

The code for volume curation (alignment and resample), hierarchical analysis, PCA-UMAP-HDBSCAN, occupancy analysis, and quadrant-dependency analysis can be found at: https://github.com/ks277/2022_50S_landscape_paper [![DOI] (https://zenodo.org/badge/563964313.svg)] (https://zenodo.org/badge/latestdoi/563964313).

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

## Acknowledgements

This work was supported by a grant from the NIH GM-136412 (to J.R.W) and NIH U54 AI170855, and the Hearst Foundations Developmental Chair (to D.L.).

## Author contributions

K.S.: Conceptualization, Investigation, Methodology, Software, Formal Analysis, Data Curation, Writing—Original Draft, Writing—Review & Editing, Visualization. N.L.: Conceptualization, Investigation, Methodology, Formal Analysis, Data Curation, Writing—Review & Editing. J.N.R.G.: Conceptualization, Investigation, Methodology, Software, Data Curation, Writing—Original Draft, Writing—Review & Editing. X.D.:Conceptualization, Investigation, Methodology, Software, Writing—Review & Editing, Visualization. D.L.: Conceptualization, Investigation, Writing—Review & Editing, Resources. J.R.W.: Conceptualization, Methodology, Software, Data Curation, Writing—Review & Editing, Visualization, Supervision, Project Administration, Funding Acquisition.

## Competing interests
The authors declare no competing interests.
