## [Peer Review File · Nature Communications]

Assembly Landscape for the Bacterial Large Ribosomal SubunitREVIEWER COMMENTS

Reviewer #1 (Remarks to the Author):

Review of Zheng et al. "Assembly Landscape..."

This paper presents an experimental and computational tour-de-force that combines biochemistry, cryo-EM and computational analysis to provide the most complete description yet of the staggeringly complex process of assembly of the 50S ribosomal subunit. Their strategy is to determine the cryo-EM structures of assembly intermediates by slowing down assembly using strains carrying mutations that affect in vivo assembly rates and low temperature, followed by a sophisticated data analysis strategy that orders and links the various intermediates in a highly detailed overall assembly scheme. Following on their earlier studies based on mutations in protein bL17 and the assembly factor SrmB, the authors expand this approach to include mutation of the assembly factor DeaD. The result is a highly self-consistent family of parallel assembly pathways that describe the order of folding of local elements of ribosomal RNA and binding of ribosomal proteins.

This is potentially a landmark paper that should be published in Nature Comm. My main recommendation to the authors is that they make a stronger effort to explain their data processing methodology in clear language that can be understood by a wide audience. In particular, much of the description of the Extended Data, which are important for a thorough understanding of the paper, is virtually incomprehensible, as are parts of the main Figures. Also, it would help to tie this work together with the early Nierhaus assembly map by presenting a Figure of the original Nierhaus map annotated to incorporate the cryo-EM analysis in some way. I realize that this may be a big ask, given the complexity of the findings, but its fundamental importance deserves the most transparent presentation. Also, there is no formal Discussion section, leaving much of the conclusions to emerge intertwined with the Results. I would encourage the authors to clarify their conclusions in a Discussion and/or Conclusions section. There is much hidden excitement.

Reviewer #2 (Remarks to the Author):

The first assembly maps of the bacterial ribosome date back to the in vitro reconstitution work from Nierhaus in the early 1980s. In the last couple decades the Williamson group has pioneered the use of newer technologies to understand ribosome assembly in vivo. To date, the studies are in overall agreement, with one of the major take homes being that the assembly pathway reflects the direction of transcription. Whereas our understanding of ribosome assembly and the specific functions of many

assembly factors is now quite well developed for eukaryotic ribosomes, our understanding of this process is less well understood in bacteria. This appears to be due in large part to the incredible speed of assembly in bacteria and the low levels of intermediates as well as the fewer number and lower dependence on assembly factors with which to fish out intermediates. In this manuscript the authors utilize deletion of DeaD, a DEAD box RNA helicase that is required at low temperature for assembly of the 50S subunit, to stall intermediates of assembly in vivo. The authors use cryo-EM to image pre-50S particles and develop novel bioinformatic tools using PCA and UMAP to tease apart and analyze the many states of pre-50S that they observe. They also compare these results to previous work from their lab, reanalyzing those data sets with their new tools. The strength of this manuscript is in the development of new bioinformatic tools for the analysis of complex heterogeneous structures. The manuscript is less compelling from a biological perspective, leaving the reader without much new insight into the mechanisms of assembly or the function of DeaD.

Specific Comments

1. The authors identify “folding blocks” of the large subunit. However, these do not correspond to discrete RNA secondary structural domains and the authors do not relate the order of folding of these blocks to known RNA folding or protein loading events. The high-level discussion of blocks without relating their function to known RNA folding leaves this discussion rather esoteric and difficult to appreciate.
2. The text is almost completely devoid of any mention of assembly factors in the structures, with the exception of a brief mention of YjgA. Were assembly factors not observed? At the least, this should be stated.
3. The work would have greater impact if the authors could derive some fundamental concepts from their work that relate to RNP assembly in general.
4. Considering the extensive analysis of eukaryotic ribosome assembly, are their points of interest in which there is commonality or divergence between bacterial and eukaryotic assembly?
5. Can the authors comment on whether or not the multiple states they observe are on-pathway or not? And, related, why did the authors choose to arrest DeaD cells at the non-permissive temperature of 19°C, rather than a semi-permissive temperature at which ribosome production is slowed but not fully arrested. The latter may be less prone to accumulation of off-pathway species.
6. It has been noted previously that DeaD associates with pre-50S at low temperature. Did the authors attempt to determine a structure of DeaD-bound pre-50S?

7. Can the authors be more specific about what is meant by pre-50S? Previous work reported that DeaD mutants accumulate both a 40S species and aberrant, presumably pre-50S, in the 50S peak. Was the pre-50S the 40S peak, aberrant 50S or everything together?

8. DeaD mutants show RNA processing defects. Is the 5'-end of 23S resolved in the various states? Can the extended 5'-ends, reported by others, be seen in the structures and does the extent of processing correlate with specific cryo-EM states?

9. How do the authors correlate their results with previous results from Charollais suggesting that DeaD normally acts at a later step than SrmB.

Reviewer #3 (Remarks to the Author):

In the present paper, Sheng and colleagues analyze the assembly of the bacterial 50s subunit. Isolating 50S precursors from a Δ deaD strain grown at low temperature and using cryo-EM they were able to obtain structural information about very early pre-50S particles. To derive a comprehensive assembly map, Sheng et al developed a novel PCA-UMAP-HDBSCAN analysis, to segment the maps and to obtain cooperative assembly blocks of ribosomal protein and elements of rRNA. Previous work mostly provided structural information about late stages of 50S assembly and the present analysis provides structural information about the early stages.

Overall, the results are interesting and provide new insights into the important assembly pathway of the large ribosomal 50S subunit. It is strange however that the authors do not mention the very recent paper by Dong et al, 2023, NAR, also from the Williamson lab, which is highly related. While both papers use different biochemical approaches to study assembly and thus stand in their own rights, the results are quite similar. Also in the previous paper, the Williamson lab reported a very early pre-50S block composed of a 600-nucleotide-long folded rRNA and three ribosomal proteins. This warrants an in-depth comparison of the previous near-physiological in vitro assembly with the present in vivo precursors.

In the previous paper, the Williamson lab used also segmentation and occupancy analysis to derive assembly blocks. What is the difference to the present method and what is the advantage of the novel PCA-UMAP-HDBSCAN method? To which extent differences are caused by the different biochemical approaches or by the different segmentation analysis?

Moreover, the present paper is hard to follow. It is partially very technical and results are presented in a rather schematic manner. It is sometimes not becoming entirely clear, if this paper is intended to describe biological result or a method. The way similar results are presented in the Dong et al paper appears more intuitive.

Additional Points:

1. State-of-the-art validation for the cryo-EM maps is missing. Resolution of the maps is hidden in Table S2 but not reported in the main text. There are no FCS plots and no local resolution estimates.
2. The cryo-EM maps are only analyzed with respect to the rRNA elements and ribosomal proteins, for which density is present. Are all elements in the same conformation as in the mature 50S subunit or are there conformational changes? If the latter is true, how the conformational changes are impacting the segmentation analysis?
3. The cryo-EM maps have to be deposited into the EMDB. At present there is a statement concerning code accessibility but no statement concerning data accessibility.
4. The authors state in text and abstract that assembly primarily proceeds in the 5'-3' direction, consistent with a co-transcriptional organization of the folding blocks. However, the uL3 block with the 3' domain VI of 23S rRNA assembles before e.g. the uL2 block and the PTC with domains IV and V. This should be clarified.
5. The nomenclature for the assembly blocks is confusing to some extent. The uL3 block (in the text) is L3 block in Fig. 3b. Here there is the L23 block, whereas in the recent paper by Dong et al. a highly similar block is named uL29 block.
6. Extended data Fig. 5 is important for understanding the results and should be incorporated into main Figs. 2 or 3 (similar to Fig. 3 of the recent paper by Dong et al.).
7. What is the rationale for basing the thresholding of maps on maximum voxel intensity and for choosing the 1% cutoff level? In cryo-EM and X-ray maps are usually scaled based on the variance / sigma level. Maximum voxel intensity may be influenced stronger by outliers. How do the chosen threshold levels of the respective maps compare to the density variance?
8. How have the molecular weight differences in kDa been derived from the thresholded volumes? How accurate is this value? To which extent may it be influenced by broadening of density due to varying flexibility of certain elements?
9. How different are the density maps from different datasets that fall into the same class? For example, in Fig. 1a/b class B-a1 from the deaD and srmB data sets, respectively, looks significantly different.

10. In Figure 1d the grey colors of the bar below the dendrogram are not explained.
11. In Fig. 3a the color code is not explained (blue squares).
12. There are 10 assembly blocks. However, Fig. 3b there are only nine blocks. Why?

Manuscript # NCOMMS-22-52807-T

"Assembly Landscape for the Bacterial Large Ribosomal Subunit"

Point-by-point response to reviewer comments, embedded below.

Reviewer #1 (Remarks to the Author):

Review of Zheng et al. "Assembly Landscape..."

This paper presents an experimental and computational tour-de-force that combines biochemistry, cryo-EM and computational analysis to provide the most complete description yet of the staggeringly complex process of assembly of the 50S ribosomal subunit. Their strategy is to determine the cryo-EM structures of assembly intermediates by slowing down assembly using strains carrying mutations that affect in vivo assembly rates and low temperature, followed by a sophisticated data analysis strategy that orders and links the various intermediates in a highly detailed overall assembly scheme. Following on their earlier studies based on mutations in protein bL17 and the assembly factor SrmB, the authors expand this approach to include mutation of the assembly factor DeaD. The result is a highly self-consistent family of parallel assembly pathways that describe the order of folding of local elements of ribosomal RNA and binding of ribosomal proteins.

This is potentially a landmark paper that should be published in Nature Comm. My main recommendation to the authors is that they make a stronger effort to explain their data processing methodology in clear language that can be understood by a wide audience. In particular, much of the description of the Extended Data, which are important for a thorough understanding of the paper, is virtually incomprehensible, as are parts of the main Figures. Also, it would help to tie this work together with the early Nierhaus assembly map by presenting a Figure of the original Nierhaus map annotated to incorporate the cryo-EM analysis in some way. I realize that this may be a big ask, given the complexity of the findings, but its fundamental importance deserves the most transparent presentation. Also, there is no formal Discussion section, leaving much of the conclusions to emerge intertwined with the Results. I would encourage the authors to clarify their conclusions in a Discussion and/or Conclusions section. There is much hidden excitement.

Response: We thank the reviewer for their enthusiasm, and for the general comments, which we have tried to address in our revisions. We agree that the Nierhaus map would be helpful, but we are already excessive in length, and a Nierhaus summary has been presented in the recent paper by Nikolay, and our own work, Dong et al. Further, the Nierhaus dependencies are actually already presented as the arrows in Figure 3, albeit in a reorganized form. We have added a discussion section to highlight and reiterate the main findings, and to discuss the relationship of our work to the previous yeast work and the

contemporaneously published bacterial work.

Reviewer #2 (Remarks to the Author):

The first assembly maps of the bacterial ribosome date back to the in vitro reconstitution work from Nierhaus in the early 1980s. In the last couple decades the Williamson group has pioneered the use of newer technologies to understand ribosome assembly in vivo. To date, the studies are in overall agreement, with one of the major take homes being that the assembly pathway reflects the direction of transcription. Whereas our understanding of ribosome assembly and the specific functions of many assembly factors is now quite well developed for eukaryotic ribosomes, our understanding of this process is less well understood in bacteria. This appears to be due in large part to the incredible speed of assembly in bacteria and the low levels of intermediates as well as the fewer number and lower dependence on assembly factors with which to fish out intermediates. In this manuscript the authors utilize deletion of DeaD, a DEAD box RNA helicase that is required at low temperature for assembly of the 50S subunit, to stall intermediates of assembly in vivo. The authors use cryo-EM to image pre-50S particles and develop novel bioinformatic tools using PCA and UMAP to tease apart and analyze the many states of pre-50S that they observe. They also compare these results to previous work from their lab, reanalyzing those data sets with their new tools. The strength of this manuscript is in the development of new bioinformatic tools for the analysis of complex heterogeneous structures. The manuscript is less compelling from a biological perspective, leaving the reader without much new insight into the mechanisms of assembly or the function of DeaD.

Specific Comments

1. The authors identify "folding blocks" of the large subunit. However, these do not correspond to discrete RNA secondary structural domains and the authors do not relate the order of folding of these blocks to known RNA folding or protein loading events. The high-level discussion of blocks without relating their function to known RNA folding leaves this discussion rather esoteric and difficult to appreciate.

Response: RNA folding in the large subunit was not well understood before this work. The secondary structure domains were annotated based on phylogenetic analysis and heuristics. Once the structure of the ribosome was solved, the folding pathway became even more obscure, as the annotated domains were all interdigitated, without any obvious way to peel back the structure in layers as a putative folding pathway. In the present work, the correlation between RNA folding and protein binding was systematically investigated using both occupancy analysis and dependency analysis on the set of observed maps. The ribosomal protein binding dependency is very consistent to what the Nierhaus group reported. The black and grey arrows in Fig.3 are actually derived from the original Nierhaus r-protein dependency. The vectorial nature of early assembly was more fully described in

our recent work on cotranscriptional ribosome assembly (Dong et al.), and also, we have analyzed the Dong's iSAT dataset together with the present three in vivo datasets, and they show consistent results.

2. The text is almost completely devoid of any mention of assembly factors in the structures, with the exception of a brief mention of YjgA. Were assembly factors not observed? At the least, this should be stated.

Response: This is in fact a major disappointment, and there is very little evidence for the presence of factors, beyond our previous observation of YjgA binding. No other assembly factors were observed in these three datasets with Cryo-EM. Some factors for example, YhbY, SrmB (not in $\Delta srmB$), DeaD (not in $\Delta deaD$) and ObgE were co-migrate with intermediate peaks which are confirmed by proteomics mass spec but we think it is hard to resolve by Cryo-EM for three reasons: 1) low abundance and occupancy by the factors in different classes, 2) binding of factors to flexible rRNA regions and 3) possible non-specific binding of to the intermediates.

One advantage of our segmentation methods is that it will tell you if there is non-native density in the dataset, for example, in bL17-depletion strain, YjgA was observed and segmented out. Unfortunately, our data are devoid of unexplained density that can be attributed to assembly factors. We note that this is in distinction to eukaryotic assembly.

We have highlighted the lack of bound factors as a distinction from the yeast work in the new Discussion section.

3. The work would have greater impact if the authors could derive some fundamental concepts from their work that relate to RNP assembly in general.

Response: Due to the highly specialized and ancient evolution of the translation apparatus, it may be difficult to generalize to other RNPs. The rRNA is highly structured, and a lot of evidence support the model of co-transcriptional assembly. Nevertheless, RNA conformational changes that are stabilized by protein binding events are likely related to formation of RNPs that regulate mRNA metabolism.

4. Considering the extensive analysis of eukaryotic ribosome assembly, are their points of interest in which there is commonality or divergence between bacterial and eukaryotic assembly?

Response: This is an excellent point that we should have amplified in the manuscript. Indeed in 60S ribosome assembly, there are nucleolar intermediates that have been identified in both the Klinge and Beckmann groups that have domains I, II and VI assembled in close analogy to our work on the 50S^{1,2}. However, there are two major distinctions. First,

we observe domain I alone, domains I/II, and domains I/III as distinct intermediates not described in the yeast work. Second, the early yeast intermediates are heavily decorated with assembly factors that are preventing further assembly, while the early bacterial intermediates are nearly devoid of assembly factors. We have added a paragraph in the manuscript to highlight these points.

5. Can the authors comment on whether or not the multiple states they observe are on-pathway or not? And, related, why did the authors choose to arrest DeaD cells at the non-permissive temperature of 19°C, rather than a semi-permissive temperature at which ribosome production is slowed but not fully arrested. The latter may be less prone to accumulation of off-pathway species.

Response: We don't specifically know the intermediates are on pathway in the $\Delta deaD$ strain. However, we did check extensively in the previous bL17-depletion paper using pulse chase experiments³, that the intermediates are competent to proceed. Given that many of the intermediates in the present work are very similar to the bL17, we think they should also be competent, but we cannot make this direct claim. Given the extraordinary importance of ribosome assembly in bacteria, it is likely that the cell will handle every eventuality, and that all of the intermediates will assemble eventually. As described in pulse labeling experiment in previous paper⁴, the pre-50S peak is chased into 70S particles, which argues the intermediate is competent to assemble. The 19 degree condition was chosen because there were more intermediates accumulated compared to higher temperatures (data not shown) which was beneficial for Cryo-EM sample preparation.

6. It has been noted previously that DeaD associates with pre-50S at low temperature. Did the authors attempt to determine a structure of DeaD-bound pre-50S?

Response: No, but that is in progress as a separate study. Also, as mentioned in question 2, there is some DeaD bound to the pre-50S peaks as judged by mass spectrometry, but the protein is either bound to the flexible rRNA regions (i.e. not resolved in our maps) or the interaction was non-specific.

7. Can the authors be more specific about what is meant by pre-50S? Previous work reported that DeaD mutants accumulate both a 40S species and aberrant, presumably pre-50S, in the 50S peak. Was the pre-50S the 40S peak, aberrant 50S or everything together?

Response: We use the Pre-50S as a general term for the ensemble of particles contains 23S rRNA but that sediments above mature 50S subunits on the sucrose gradient.

8. DeaD mutants show RNA processing defects. Is the 5'-end of 23S resolved in the various states? Can the extended 5'-ends, reported by others, be seen in the structures and does the

extent of processing correlate with specific cryo-EM states?

Response: We have seen this in several other studies, but we do not see evidence in our maps for this, likely due to the relatively low resolution and the small size of the RNA processing remnants.

9. How do the authors correlate their results with previous results from Charollais suggesting that DeaD normally acts at a later step than SrmB.

Response: This is a complex subject. The specific steps facilitated by either DeaD or SrmB are not clear, and since these earlier studies, other non-ribosomal roles have been implicated for these helicases. We know that there are parallel assembly routes for the 50S subunit, that neither helicase is essential, and that both facilitate assembly at lower temperatures. We do not think that every assembly channel requires helicase assistance, and it is difficult to disentangle the folding problems associated with cold from those associated with lack of helicase. We lack specific information about what the actual substrates are for any of the helicases, and as yet, there is no structural information localizing the helicases. In sum, we are hopeful that the copious literature on helicases will be reconciled mechanistically, at some point in the future. While the $\Delta deaD$ strain provided us with tremendous insights into cooperative folding domains in assembly, we unfortunately did not get any insights into the specific mechanism of DeaD in assembly.

Reviewer #3 (Remarks to the Author):

In the present paper, Sheng and colleagues analyze the assembly of the bacterial 50S subunit. Isolating 50S precursors from a $\Delta deaD$ strain grown at low temperature and using cryo-EM they were able to obtain structural information about very early pre-50S particles. To derive a comprehensive assembly map, Sheng et al developed a novel PCA-UMAP-HDBSCAN analysis, to segment the maps and to obtain cooperative assembly blocks of ribosomal protein and elements of rRNA. Previous work mostly provided structural information about late stages of 50S assembly and the present analysis provides structural information about the early stages.

Overall, the results are interesting and provide new insights into the important assembly pathway of the large ribosomal 50S subunit. It is strange however that the authors do not mention the very recent paper by Dong et al, 2023, NAR, also from the Williamson lab, which is highly related. While both papers use different biochemical approaches to study assembly and thus stand in their own rights, the results are quite similar. Also in the previous paper, the Williamson lab reported a very early pre-50S block composed of a 600-nucleotide-long folded rRNA and three ribosomal proteins. This warrants an in-depth comparison of the previous near-physiological in vitro assembly with the present in vivo precursors.

Response: The Dong 2023 paper was not mentioned in the manuscript due to the complexities and timing of peer review for the two manuscripts. They were written contemporaneously but submitted independently, and our original intent was for the present work to be published first. The peer review and revision process at *NAR* was extremely rapid, while the present manuscript worked its way through the *Nature* system on a longer trajectory. No matter, now that Dong is published, we can refer to it and compare with the present work. The Dong paper used our previously described iterative subclassification method, and occupancy analysis to deduce the folding blocks^{3,5,6}, and the present manuscript is indeed the first application of the novel PCA-UMAP-HDBSCAN method on completely separate datasets.

In the previous paper, the Williamson lab used also segmentation and occupancy analysis to derive assembly blocks. What is the difference to the present method and what is the advantage of the novel PCA-UMAP-HDBSCAN method? To which extent differences are caused by the different biochemical approaches or by the different segmentation analysis?

Response: The old structural elements are defined by native structure form 70S and an arbitrary segmentation of RNA helices. The present method makes no assumption of what the segments are and can capture elements that are not present in the native structure and had finer definition of RNA helices.

We feel that the differences observed are due to both the biochemical approaches and the analysis methods. As we describe in the paper and as the reviewer points out, our previous work used a fairly arbitrary definition of structural units based on the r-proteins and numbered helices in the rRNA secondary structure. The new segmentation method does not require any assumptions about the folding elements, rather, they emerge from the data analysis, making it generally applicable to any set of maps in any system. In addition, the new method implicitly handles “non-native density”, while the old method can only detect density in the defined regions. A particularly striking case is helix 38, which was naturally divided into two regions by the new method, and the observation of non-native density corresponding to YjgA.

Furthermore, in this manuscript we demonstrate that the segments and dependencies identified in the DeaD dataset serve as a framework, or landscape, that supports interpretation of datasets not included in the analysis. The set of dependencies we observe allow for combinations of segments that were not present in the DeaD dataset, but that were observed in others (SrmB, bL17). This is a strong validation of the segmentation and dependency analysis, that expands significantly and more generally over the iSAT work reported in Dong.

Moreover, the present paper is hard to follow. It is partially very technical and results are presented in a rather schematic manner. It is sometimes not becoming entirely clear, if this paper is intended to describe biological result or a method. The way similar results are presented in the Dong et al paper appears more intuitive.

Response: The reviewer has succinctly captured a challenge for the manuscript. It is like a house-boat, which is not a particularly comfortable home, nor a particularly effective boat. Any given reader (reviewer) might be more interested in the house aspects or the boat aspects. But a house-boat is actually a novel structure/craft with unique features. This manuscript is both a method, and a biological result, and the impact of each aspect is enhanced synergistically.

Additional Points:

1. State-of-the-art validation for the cryo-EM maps is missing. Resolution of the maps is hidden in Table S2 but not reported in the main text. There are no FCS plots and no local resolution estimates.

Response: The FCS curves have all been calculated and deposited with the maps in the EMDB (not an easy task for a set of maps of this size), and the ID has been added to the supplementary table. We are working on binned data with maximal 5.24 Å resolution (Nyquist). Our conclusions are largely based on the gross features of the maps, and we do not think that local resolution is necessary or helpful in this study.

2. The cryo-EM maps are only analyzed with respect to the rRNA elements and ribosomal proteins, for which density is present. Are all elements in the same conformation as in the mature 50S subunit or are there conformational changes? If the latter is true, how the conformational changes are impacting the segmentation analysis?

Response: The unbiased segmentation will reveal any areas that have undergone conformational changes, thus appearing as a distinct segment in a set of maps. changes. For example, in one of our other datasets not included in this work, we find a folding block that has a mis-docked CP. A continuous conformational ensemble cannot be readily discerned using our segmentation analysis, primarily because conventional reconstruction does not resolve distinct states in the presence of a continuum. We focused on intermediate states in which the helices that could be reconstructed are either trapped in an intermediate state or in the mature state.

3. The cryo-EM maps have to be deposited into the EMDB. At present there is a statement concerning code accessibility but no statement concerning data accessibility.

Response: All maps have been deposited and validated through EMDB for release upon publication. The EMDB IDs have now been included in the supplementary table.

4. The authors state in text and abstract that assembly primarily proceeds in the 5'-3' direction, consistent with a co-transcriptional organization of the folding blocks. However, the uL3 block with the 3' domain VI of 23S rRNA assembles before e.g. the uL2 block and the PTC with domains IV and V. This should be clarified.

Response: What we observe is generally aligned the co-transcriptional model, but as the reviewer notes, domain VI can dock prior to domains IV/V. This is actually similar to what has been previously observed in yeast and human nucleolar intermediates, and we have elaborated on this in the new Discussion section. In addition to the co-transcriptional trend, the solvent side of the subunit is formed before the inter-subunit side. We proposed a scaffold model in this paper, in which the solvent side of 50S generally follows the co-transcriptional order and forms a scaffold for the other parts including PTC to build on.

5. The nomenclature for the assembly blocks is confusing to some extent. The uL3 block (in the text) is L3 block in Fig. 3b. Here there is the L23 block, whereas in the recent paper by Dong et al. a highly similar block is named uL29 block.

Response: We agree that the nomenclature can be confusing. We will ensure consistent naming within the manuscript. The names between datasets can be different due to the different segmentation.

6. Extended data Fig. 5 is important for understanding the results and should be incorporated into main Figs. 2 or 3 (similar to Fig. 3 of the recent paper by Dong et al.).

Response: We agree the secondary structure is important to present, and we will incorporate it into Figure 2.

7. What is the rationale for basing the thresholding of maps on maximum voxel intensity and for choosing the 1% cutoff level? In cryo-EM and X-ray maps are usually scaled based on the variance / sigma level. Maximum voxel intensity may be influenced stronger by outliers. How do the chosen threshold levels of the respective maps compare to the density variance?

Response: We thank the reviewer for catching this, because we mis-stated the thresholding method in the manuscript.

First, in our previous paper ⁵, we did in fact use a 3*sigma noise cutoff to threshold unmasked and unsharpened maps. In the present work we work with masked and sharpened maps for clarity, and the noise-based cutoff is not appropriate.

To the reviewers point, we did not threshold at 1% of the maximum intensity, but rather took the top 1-percentile of voxels ranked by intensity for each individual map. In this case the actual threshold value is relatively insensitive to a small number of outliers, and we visually inspected each thresholded map in Chimera.

Further, the thresholding was used as the basis for identifying a *consensus* set of voxels in the set of 21 maps with significant intensity in at least one map. The entire consensus set of voxels was used from each individual map. Our box size was 160x160x160, and 1% is 40,960 voxels for a given map. The consensus set of voxels is 114,392, meaning a significant number of "subthreshold" voxels were included from any individual map in the global analysis. Thus, there is effectively no threshold for any individual map, but rather we globally define the region of interest. We did this because the maps differed greatly in the number of particles in the classes. Many consensus voxels have significant intensity in some maps, but "noise" in others. This is in fact the basis for discriminating segments in the PCA-UMP-HDBSCAN approach. We have restated the methods, and we regret the error and confusion.

8. How have the molecular weight differences in kDa been derived from the thresholded volumes? How accurate is this value? To which extent may it be influenced by broadening of density due to varying flexibility of certain elements?

Response: The molecular weight differences are essentially a voxel count scaled by the voxel size and a density. The values are certainly affected by the variations in voxel intensities. Operationally, we tend to ignore differences that are less than "10 kDa", and we don't view them as particularly accurate, but rather as a rough guide to the size difference on an intuitive scale.

9. How different are the density maps from different datasets that fall into the same class? For example, in Fig. 1a/b class B-a1 from the deaD and srmB data sets, respectively, looks significantly different.

Response: We regret this confusion, and we have clarified the figure legend. The classes are numbered independently in the three datasets. The full designation would be deaD: B-a1 and srmB: B-a1 (as for the labels in Figure 1d). The difference between these two classes can be understood in the dendrogram of Figure 1d. The dendrogram has major classes preB, B, C, D, E, G, and B,C,D,E are subdivided into a and b subclasses. Particles deaD: B-a1 and srmB: B-a1 are both members of the B-a subclass, but they are deeply branched within the subclass, consistent with the reviewer's visual observation. The depth corresponds roughly to a molecular weight difference, calculated as the Euclidean distance between the

maps (volume of the difference map). This is another reason to use MW differences, as it makes the y-axis on the dendrogram more interpretable.

10. In Figure 1d the grey colors of the bar below the dendrogram are not explained.

Response: The grey colors are a key to which dataset the class came from, corresponding to the bars at the top of Fig 1a,b,c: black = *deaD*, dark gray = *srmB*, light gray = *bL17*, and this has been added to the legend.

11. In Fig. 3a the color code is not explained (blue squares).

Response: The blue color intensity ranges from white (no occupancy) to blue (full occupancy), and this has been added to the figure legend.

12. There are 10 assembly blocks. However, Fig. 3b there are only nine blocks. Why?

Response: One of the blocks (*uL1*) appeared in one class only in the Δ *deaD* dataset. We felt that the dependencies of this block would be entirely assigned based a single intermediate, which was not well-supported. A goal of ongoing other work is to better understand the dependencies for the *uL1* block.

- 1 Kater, L. *et al.* Visualizing the Assembly Pathway of Nucleolar Pre-60S Ribosomes. *Cell* **171**, 1599-1610 e1514 (2017). <https://doi.org:10.1016/j.cell.2017.11.039>
- 2 Sanghai, Z. A. *et al.* Modular assembly of the nucleolar pre-60S ribosomal subunit. *Nature* **556**, 126-129 (2018). <https://doi.org:10.1038/nature26156>
- 3 Davis, J. H. *et al.* Modular Assembly of the Bacterial Large Ribosomal Subunit. *Cell* **167**, 1610-1622 e1615 (2016). <https://doi.org:10.1016/j.cell.2016.11.020>
- 4 Peil, L., Virumäe, K. & Remme, J. Ribosome assembly in *Escherichia coli* strains lacking the RNA helicase *DeaD/CsdA* or *DbpA*. *Febs j* **275**, 3772-3782 (2008). <https://doi.org:10.1111/j.1742-4658.2008.06523.x>
- 5 Rabuck-Gibbons, J. N., Lyumkis, D. & Williamson, J. R. Quantitative mining of compositional heterogeneity in cryo-EM datasets of ribosome assembly intermediates. *Structure* **30**, 498-509 e494 (2022). <https://doi.org:10.1016/j.str.2021.12.005>
- 6 Dong, X. *et al.* Near-physiological in vitro assembly of 50S ribosomes involves parallel pathways. *Nucleic Acids Res* **51**, 2862-2876 (2023). <https://doi.org:10.1093/nar/gkad082>

REVIEWERS' COMMENTS

Reviewer #2 (Remarks to the Author):

The authors have addressed my questions and concerns. I recommend accepting the manuscript. Its strength is in providing novel methodology for analyzing complex structural data sets.

Reviewer #3 (Remarks to the Author):

The authors have adequately answered all questions and have done a great job improving the paper. It is still no easy to read, but given the enormous complexity of the subject, there is probably not much that could be done.